
# New Particle Formation and Sub-10 nm Size Distribution Measurements during the A-LIFE field experiment in Paphos, Cyprus

Sophia Brilke[1], Nikolaus Fölker[1], Thomas Müller[2], Konrad Kandler[3], Xianda Gong[2], Jeff Peischl[4,5], Bernadett Weinzierl[1] and Paul M. Winkler[1]

[1]Faculty of Physics, University of Vienna, Vienna, 1090, Austria
[2]Leibniz Institute for Tropospheric Research e.V. (TROPOS), Leipzig, 04318, Germany
[3]Technical University Darmstadt, Darmstadt, 64287, Germany
[4]Cooperative Institute for Research in Environmental Studies, University of Colorado Boulder, Boulder, CO 80309, USA
[5]NOAA Earth System Research Laboratory, Chemical Science Division, Boulder, CO 80305, USA

*Correspondence to*: Sophia Brilke (sophia.brilke@univie.ac.at)

**Abstract.** Atmospheric particle size distributions were measured in Paphos, Cyprus, during the A-LIFE (**A**bsorbing aerosol layers in a changing climate: aging, **life**time and dynamics) field experiment from April 3 – 30, 2017. The newly developed DMA-train is deployed for the first time in an atmospheric environment for the direct measurement of the nucleation mode size range between 1.8 – 10 nm diameter. The DMA-train setup consists of seven size channels, of which five are set to fixed particle mobility diameters and two additional diameters are obtained by alternating voltage settings in one DMA every 10 s. In combination with a conventional Mobility Particle Size Spectrometer (MPSS) and an Aerodynamic Particle Sizer (APS) the complete atmospheric aerosol size distribution from 1.8 nm – 10 µm is covered. The focus of the A-LIFE study is to characterize new particle formation (NPF) in the Eastern Mediterranean region at a measurement site with strong local pollution sources. The nearby Paphos airport was found to be a large emission source for nucleation mode particles and we analysed the size distribution of the airport emission plumes at approximately 500 m from the main runway. The analysis yielded 9 NPF events in 27 measurement days from the combined analysis of the DMA-train, MPSS and trace gas monitors. Growth rate calculations were performed and a size-dependency of the initial growth rate (< 10 nm) is observed for one event case. Fast changes of the sub-10 nm size distribution on the timescale of a few minutes are captured by the DMA-train measurement during early particle growth and are discussed in a second event case. In 2 cases, particle formation and growth were detected in the nucleation mode size range which did not exceed the 10 nm threshold. This finding implies that NPF likely occurs more frequently than estimated from studies where the lower nanometre size regime is not covered by the size distribution measurements.





## 1 Introduction

New particle formation (NPF) is a major source of nucleation mode particles which form from low-volatility vapours in the atmosphere. Freshly formed particles can eventually grow to sizes where they scatter light or become cloud condensation nuclei (CCN). NPF is known to significantly contribute to the global total particle concentration and CCN budget (Merikanto et al., 2009; Spracklen et al., 2006). Understanding the feedback on the global climate of secondary aerosol requires a profound knowledge of the NPF frequency and processes promoting the formation and growth. Global modelling studies

need a large set of observation points throughout the atmosphere. The number of sites observing NPF is increasing and includes studies in remote areas (Kecorius et al., 2019; Pikridas et al., 2012; Weber et al., 1995), coastal and continental regions (Hoffmann et al., 2001; Mäkelä et al., 1997; Weber et al., 1997), the free troposphere (Bianchi et al., 2016; Williamson et al., 2019) and polluted environments (Hamed et al., 2007; Stolzenburg et al., 2005; Yao and Garmash et al., 2018; Yu et al., 2016).

Size-resolved particle measurements down to the diameters of freshly nucleated particles provide insight into the dynamics of the processes of aerosol nucleation and growth. Commonly, measurements are performed by charging particles using a radioactive or x-ray ionization source, classifying them according to their electrical mobility using a Differential Mobility Analyser (DMA) and detecting them in a Condensation Particle Counter (CPC) or Faraday Cup Electrometer (FCE). Measuring size-resolved particle number concentrations in the nucleation mode size range is challenging because the

efficiencies for transmission, charging and detection of sub-10 nm particles are low.

Different approaches have been made to access information on the dynamics of nucleation mode particles. The Neutral cluster and Air Ion Spectrometer (NAIS) is a frequently implemented method to measure the size distribution of charged and neutral clusters (< 3 nm) using a DMA coupled to a set of parallel FCEs (Manninen et al., 2009). The advantage of using the FCE as a detector is that it provides a size-independent detection of charged particles whereas the drawback lies in the fact

that FCEs have a lower sensitivity compared to CPCs. Therefore, higher particle concentrations must be present to achieve a measurable signal. Mobility Particle Size Spectrometers in scanning mode operation (MPSS) commonly classify particles in a DMA and optically detect them in a CPC. The sensitivity of the CPC to low particle concentrations is higher compared to FCEs but the size-dependent detection of the used CPC results in larger uncertainties, requiring detailed calibration procedures (Kangasluoma and Kontkanen, 2017). By implementing a low cut-off CPC, e.g. a diethylene glycol-based CPC,

the complete number size distribution of neutral particles has been measured during nucleation events in the atmosphere (Jiang et al., 2011). In recent efforts, the Caltech nano-Scanning Electrical Mobility Spectrometer (nSEMS) was developed to combine the measurement of neutral particles and particles of both polarities in the lower nanometre size regime with minimised diffusional losses (Amanatidis et al., 2019). However, scanning the voltage of a DMA over the complete mobility range takes time and the time resolution of a scanning device typically varies between 5 to 10 minutes. Counting statistics

become an issue when the signal in the detector is low which is the case for sub-10 nm particles due to large diffusional losses despite high number concentrations during nucleation events.



In this study, we meet the challenge of measuring sub-10 nm particles using the newly-developed DMA-train (Stolzenburg et al., 2017). In contrast to conventional particle sizers with a single scanning classifier, the DMA-train utilizes six DMAs in parallel operated at fixed voltages with each DMA coupled to a CPC. This arrangement allows size distribution
measurements of neutral atmospheric particles at a time resolution on the order of a few seconds. The main advancement is that the DMA-train has an increased sensitivity towards low particle counts especially in the lower size channels, due to the permanent monitoring of individual sizes that can be used for statistical analysis. During the A-LIFE field experiment in Cyprus, the DMA-train was, for the first time, implemented in an atmospheric environment. The Mediterranean region is subject of various aerosol types from different sources such as biomass burning, fuel combustion, sea spray aerosol, mineral
dust and secondary aerosol. Here, we focus on secondary aerosol in the Paphos region where strong local pollution sources are present, and capture the sub-10 nm particle dynamics during nucleation events. NPF was found to frequently occur in the Eastern Mediterranean atmosphere (Debevec et al., 2018; Kalivitis et al., 2008, 2015; Manninen et al., 2010; Pikridas et al., 2012), and the results from this study contribute more detailed information on the nucleation mode particle size range.

## 2 Methods

### 2.1 Sampling site

The island of Cyprus is situated in the Eastern Mediterranean Sea (Figure 1) and is a unique location where air masses converge from several distinct origins carrying complex aerosol mixtures. The Sahara Desert and the Arabian Peninsula are sources of mineral dust-rich air masses from the south. Contributions from biomass burning are expected to originate from the Turkish mainland to the northeast. Anthropogenic pollution is transported to the Cyprus region from Southeast Europe
and the Middle East. The Paphos ground-based station (34°42'40.5"N 32°28'58.5"E) is close to the Paphos International Airport, which is located approximately 500 m to the northeast from the measurement site. The coastline is 50 m away and extends from the southeast to the northwest. Emissions from ships arriving at and passing by Paphos are expected to contribute to the local urban pollution. The nearest and largest urban center is Paphos, with 36000 inhabitants located 5 km to the northwest. The streets surrounding the measurement station carry a medium-to-high traffic load. The Troodos
mountain range, located about 30 km from the station, contains a large amount of vegetation and reaches up to 2000 m elevation. In summary, the sampling site can be characterized as rural polluted with a marine background and strong local pollution sources.

In Figure 1, the wind roses in the right panel show the local land-sea-breeze system with the prevailing wind direction from the northeast during night-time (18:00 – 06:00) at low wind speeds, and from the northwest during day-time  (06:00 – 18:00)
at higher wind speeds.



## 2.2 Ground-based measurements

The A-LIFE intensive measurement campaign was conducted from April 3 – 30, 2017. Measurements of the particle size distribution were performed using a DMA-train in the size range between 1.8 - 10 nm at a time resolution of 1 s. A MPSS (Wiedensohler et al., 2018) in combination with an Aerodynamic Particle Sizer (APS) covered the size range from 10 nm –

10 μm at 5 min time resolution. An aerosol $PM_{10}$ inlet was installed on top of the measurement container to remove particles larger than 10 μm in aerodynamic diameter (find a more detailed description in Gong et al. (2019)). The total particle concentration was measured at 1 s time resolution using an Airmodus A10 Particle Size Magnifier (PSM) in combination with a TSI UCPC Model 3776 as a detector with a dried total inlet flow rate of 10 L/min using a core sampling probe (RH < 40%). Complementary trace gas measurements were performed using a $NO$-$NO_2$ monitor, $O_3$ monitor and $SO_2$ monitor with

periodic background measurements performed using synthetic air. The $NO$-$NO_2$ monitor was calibrated after the measurement campaign. $SO_2$ data are available for the last third of the measurement period without calibration; therefore, we use the $SO_2$ data qualitatively. Meteorological parameters such as temperature, relative humidity, solar irradiance, wind speed and wind direction were measured throughout the measurement period.

## 2.3 The DMA-train experimental setup

The DMA-train measures the particle size distribution starting at 1.8 nm mobility diameter up to 10 nm in seven size channels. The setup is presented schematically in Figure S1 in the supplemental information (SI). The sample aerosol is drawn through 1" stainless-steel tubing at a total flow rate of 21 L/min. A core sampling probe extracts a flow of 11 L/min to the DMA-train system consisting of two symmetric layers with an aerosol charger and three DMAs each followed by a CPC. The sample flow is evenly split into two 5.5 L/min flows which are guided to one layer each. The aerosol sample enters a

soft x-ray based Advanced Aerosol Neutralizer (TSI 3088, AAN) where particles are charged for the subsequent classification in three DMAs (Grimm S-DMAs) prior to particle detection in the particle counters. The aerosol flow rate in the DMAs is determined by the inlet flow rate of the CPCs. Each DMA is operated at a fixed set voltage and particles of the respective mobility diameter are counted in each size channel. The DMA-train is a modular system where single parts, e.g. the CPCs, can be easily exchanged depending on the requirements of a measurement. Here, an Airmodus A10 Particle Size

Magnifier (PSM), a turbulent mixing-type DEG-based growth stage, was implemented at the first and lowest channel for the activation of the smallest particles with a butanol-based TSI CPC Model 3772 for particle detection at a total inlet flow rate of 2.5 L/min. The second channel was equipped with the laminar-flow DEG-based TSI UCPC Model 3777 ("nanoEnhancer") followed by a butanol-based TSI UCPC Model 3772, also at a flow rate of 2.5 L/min. In the third and fourth channels, two TSI UCPC Model 3776 with an inlet flow rate of 1.5 L/min were implemented and operated at modified

temperature settings, i.e. the saturator temperature was set to 30.1°C and the condenser temperature set at 0.1°C for enhanced particle activation (Barmpounis et al., 2018; Tauber et al., 2019). A water-based TSI CPC Model 3788 was employed in the





fifth channel. The last channel was operated with a TSI UCPC Model 3776 at standard temperature settings (see Table S1 in SI). Here, the DMA voltage was not entirely fixed but alternated in a 10 s time interval to cover one additional size channel. The DMAs are operated using a closed-loop sheath flow rate of 15 L/min. Silica gel dryers are used in every sheath flow
loop to keep the RH of the sheath flow below 10%.

### 2.4 DMA-train data analysis

#### 2.4.1 DMA-train data considerations

The inversion of the DMA-train data to get information on the number concentration of particles in each size bin at each single DMA is performed as described in Stolzenburg and McMurry (2008). Detailed information on the instrument
transmission of the DMA-train setup and transfer function of the DMAs can be found in Stolzenburg et al. (2017). The particle charging efficiency is calculated from Wiedensohler's approximation (Wiedensohler, 1988) for negatively charged particles. However, the counting efficiency of the CPC renders a large uncertainty when the classified diameter lies in the size range where the diameter-dependent counting efficiency is increasing, i.e. in the ascent of the counting efficiency curve. This is especially true for the smallest classified particle sizes of the DMA-train setup. The CPC counting efficiency is
dependent on the aerosol composition, seed solubility, the RH of the sample flow and instrument settings. Here, we account for the possible above-mentioned effects on the counting efficiency when the aerosol sample is exposed to humid air. Previous studies have shown that the RH and the seed solubility with respect to the working fluid have an influence on the CPC detection efficiency (Kangasluoma et al., 2013; Tauber et al., 2019). To appropriately account for the seed variability present in the atmosphere, we used the mean counting efficiency at the classified diameter using results of the counting
efficiency for silver and sodium chloride particles exposed to 0% or 10% RH as a best estimate. The range of RH of the aerosol flow at the CPC inlets between 0 and 10% is limited by the controlled humidity of the sheath flows of the DMAs. The uncertainty on the counting efficiency from the mentioned factors is estimated to $\pm 10\%$. As an overall estimate we assume $\pm 20\%$ uncertainty on the number concentration in each size channel.

Background measurements were conducted at least twice a day with the DMA voltage in each channel set to zero. The
background values were typically 0 cm$^{-3}$ except for the Airmodus PSM, which varied with the outside RH between 0 and 1 cm$^{-3}$. The DMA-train's increased sensitivity to a low particle signal is dependent on a stable background signal. The PSM instrument settings have been adjusted accordingly, and the possible effects of this correction were considered during the analysis.

#### 2.4.1 Growth rate analysis

Detailed growth rate measurements using the appearance time method have previously been performed using the DMA-train in chamber studies (Stolzenburg et al., 2018). During ambient measurements, the growth rate measurement is not always as obvious as during laboratory studies. Inhomogeneous atmospheric mixing and changing atmospheric parameters may affect





the number concentration measured in each size channel and lead to a less distinct signal used for the growth rate analysis. A method for determining the particle growth rate based on the evolution of the nucleation mode peak was presented by

Lehtinen and Kulmala (2003). In this study, the time, $t_{mode}$, of the maximum of each particle size channel, i.e. $D_{p,mode}$, was determined using a Gaussian fit which is demonstrated to work well in Figure S2. As a best estimate for the uncertainty of the time of the mode peak, the full-width-half-maximum of each fit was used. The uncertainty on the diameter is given by the DMA flow configuration.

The growth rate (GR) is calculated from the offset between the mode diameter, $\Delta D_{p,mode}$, and the times of the peak mode,

$\Delta t_{mode}$:

$$\mathrm{GR} = \frac{\Delta D_{p,mode}}{\Delta t_{mode}}.$$

(1)

In this analysis, the mode diameter is plotted versus the time of the mode peak as presented in Figure S3. A linear fit from an orthogonal-distance-regression of the data is used to determine the growth rate directly, and accounts for the error in the time of the mode peak and the mode diameter. This method was applied in two size intervals, i.e. between 1.8 – 3.2 nm and 3.2 – 10 nm, where differences in the growth rate have been observed.

### 2.4.3 Event classification

One objective of this study is the characterization of the measurement site regarding the occurrence of NPF. An NPF event is determined from the nanoparticle size distribution data. The classification scheme for sorting a measurement day of differential mobility particle sizer (DMPS) data into the three categories, event, non-event and undefined day was first introduced by Dal Maso et al. (2005). In this established classification scheme, a day is classified into an event day if an NPF

event takes place based on visually analysing the size distribution data. The criteria for an NPF event to take place are given by the appearance of a new particle mode (< 25 nm), which persists at least for one hour and grows within several hours. Mazon et al. (2009) added classification criteria to categorize undefined days into sub-categories such as pollution-related peaks and ultrafine-mode peaks.

The criteria given by Dal Maso et al. (2005) were applied to the MPSS data and adjusted for categorizing the DMA-train

data, i.e. the sub-10 nm nanoparticle population, into event, non-event and undefined days as displayed in Figure 2. The DMA-train data was visually analysed and classified into the three cases. Trace gas data were included in the NPF analysis to further distinguish between pollution-related peaks and NPF events. Accordingly, the MPSS size distribution data were analysed for NPF event days based on the classification scheme of Dal Maso et al. (2005). In the following sections, specific event cases of particle nucleation and growth that were observed during the A-LIFE measurement period are presented and

discussed.



## 3 Results and discussion

### 3.1 New particle formation events during A-LIFE

Several events were classified based on the DMA-train classification scheme depicted in Figure 2 in the sub-10 nm size regime. Figure 3 shows three typical cases of the resulting DMA-train size distribution illustrating an event, non-event and undefined day. Panel (a) shows a size distribution revealing a new mode of nanoparticle growth which is classified as an event day. Panel (b) demonstrates a non-event case. When a mode of nanoparticles appears but does not continue to grow, it is referred to as an undefined day, which is shown in the third panel (c). Table 1 summarizes the days when an event was either observed in the DMA-train or in the MPSS particle size distribution. The remaining days were classified as non-event days. Additionally, results of the condensation sink (CS), which is the rate at which molecules condense onto pre-existing particles, were added (Lehtinen et al., 2003). For the sub-10 nm size distribution, an event was identified in 5 cases whereas 7 event days were found for particles larger than 10 nm during 27 total measurement days. The events were observed during morning hours after sunrise when solar radiation and photochemistry typically trigger the formation of clusters. The combined analysis of the sub-10 nm particle size distribution obtained by the DMA-train and the particle population between 10 – 1000 nm from the MPSS data shows that an event does not necessarily occur both in the DMA-train and the MPSS. However, the non-coincidence of events can have several reasons: i) an NPF event occurs but particles do not grow to sizes large enough to be detected in the MPSS, e.g. because of a lack of condensable vapours, ii) the particle concentration is below the DMA-train's lower detection limit or an existing particle population is carried to the measurement station with a mode diameter larger than 10 nm or, iii) meteorological conditions change, e.g. a change in air mass or wind direction occurs. We have analysed the particle size distribution data under consideration of the above-mentioned factors. Regarding the first possibility, i), the condensation sink, as listed in Table 1, does not give an indication. The reason for a lack of event identification in the DMA-train on April 8 and 9 is likely provided by the explanation given in ii) by an air mass change. On April 8 a night-time NPF event at particle concentrations lower than during the typical day-time event occurs, which suggests that the signal was insufficient for detection in the DMA-train. It is also possible, that the first steps of particle formation happened at another location and the particle population was transported to the measurement station. Interestingly, two days (April 16 and 20) show a growing mode in the sub-10 nm regime which does not continue to grow. The analysis of the meteorological parameters showed that at the stage of early particle growth the wind direction changed. The switch from land breeze to sea breeze typically occurs during morning hours, which here happened just after the start of early particle growth. Evidently, the conditions promoting the growth may change which leads to a disruption of the growing particle mode. Likewise, the particle population may be carried away and evolve at some other location. The important fact here is that NPF might occur more frequently than is assumed from studies that do not include the size distribution below 10 nm. In the following, the event case of April 16 will be discussed in more detail.

During the time window of the A-LIFE campaign, nucleation mode particles were found to originate either from NPF or strong pollution sources, such as the nearby airport.


**3.2 Nucleation mode particles from new particle formation and growth rates**

We have presented a scheme to classify events, non-events and undefined sequences of the time series of the number size distribution. Two event cases are discussed here regarding their correlation with trace gases, meteorological parameters and particle growth. The first event, on April 16, shows a clear growing nucleation mode which is suppressed before it is observed by the instrumentation covering particle sizes larger than 10 nm. In the second case, an NPF event is detected with a mode diameter growing from the lowest detectable sizes into the Aitken mode throughout the whole day of April 22.

**3.2.1 April 16**

On April 16, 2017, a growing nucleation mode was detected in the DMA-train during morning hours and is presented in Figure 4. The mode diameter is shown as white markers. The times of the mode maxima were determined from the time series of each size channel (see Figure S2 for more details). Note that in Figure 4, unconventionally, the y-axis is linear and not logarithmic. Clearly, the initial particle growth below 3.2 nm is faster compared to the larger size intervals. For the

smallest particles the growth rate is 18.2 nm/h, which is then reduced to 5.5 nm/h. The meteorological data indicate a wind direction change during the event from northeast to southeast (see Table 1). A new air mass being carried to the measurement station might lead to dilution of the condensable vapours participating in the sub-10 nm growth.

From Figure 4 it is seen that during this NPF event case the growth of freshly formed particles stops below 10 nm. Kalivitis et al. (2008) found a depletion of young Aitken mode particles and a low overall number of event cases during an NPF study

in Finokalia, Crete. The authors conclude that either the production of particles by NPF is insignificant or that particles are scavenged before reaching their observational size limit of 18 nm. From the results of our study, we conclude that a significant number of sub-10 nm particles can be formed by NPF but may not be discovered at larger sizes for various reasons. Thus, the observational size limit for NPF should be lowered to the lowest detectable sizes that can be detected with state-of-the-art instrumentation, i.e. 1 – 2 nm. We suggest that NPF frequency and the production of new particles is

underestimated when the nucleation mode size range is not completely covered.

The full atmospheric size distribution at the time of the appearance of the nucleation mode averaged during a time interval of 2 minutes is shown in panel (a) of Figure 5. The different modes are highlighted with a clear nucleation mode (red) exhibiting the highest number concentrations followed by the Aitken mode (blue), accumulation mode (green) and coarse mode (yellow). Panel (b) of Figure 5 shows the nucleation mode has begun fading out, and by panel (c) has completely

disappeared, while the Aitken, accumulation and coarse modes remain. This case study demonstrates that the gap in the size distribution of neutral atmospheric nucleation mode particles is well captured by the DMA-train which is oftentimes not the case when conventional instrumentation is deployed. The transition from nucleation to Aitken mode is not fully resolved here (see Figure 5). However, the need to combine instrumentation capable of resolving the complete size range becomes evident such that nucleation and Aitken modes are both well-covered.



### 3.2.2 April 22

An NPF event that is characterized by particle growth throughout a whole day is presented in Figure 6. Wind direction (grey markers) and total particle concentration (light green), shown in panel (a), correlate well with the trace gas concentrations shown in panel (b). The low NO and NO2 concentrations (dark green and red traces, respectively) at the start time of the event indicate that the large increase in the number concentration of particles at sizes between 1 – 4 nm is free of plume impact from local sources. Panel (c) shows the time series of the size distribution where the black markers indicate the mode peak diameter. A change in wind direction during the early particle growth from northeast to northwest is marked by the black dashed line. Despite the wind direction change, the mode continues to grow with a slight interruption due to a wind fluctuation at around 08:00. With wind coming from the northwest, i.e. from the Paphos city center and the sea, elevated concentrations of SO2 are carried to the measurement station, indicating that ship emissions may be a possible source. Panel (d) shows a close-up of the start of the event in the sub-10 nm size range. It should be noted that a burst-like behaviour is observed in the lowest size channels with an increasing mode diameter in two stages. A new nucleation mode appears for a time length of 10 minutes, disappears and returns for another 15 minutes having a larger mode diameter. The simultaneous increase of the channels during each burst makes the time-offset between the single channels indiscernible. Therefore, we did not determine a growth rate for the time window of the burst-like behaviour. A dynamic process is observed during the early particle growth which, in this case, cannot be explained by peaks in trace gas concentrations or meteorological parameters (clear sky conditions, no change in air mass). However, inhomogeneous mixing of the atmospheric sample volume can disturb the picture during ambient measurements. The growth rate of the higher size interval is 5.8 nm/h and is similar to the growth rate on April 16 and 20 (5.5 and 5.7 nm/h, listed in Table 1). On April 22, particles continue growing to sizes where they can potentially act as CCN, which was shown in the A-LIFE study on CCN by Gong et al. (2019).

### 3.3 Nucleation mode particles from airport emission plumes

A significant source contributing to high nanoparticle concentrations was found to be emissions from the nearby airport. Here, all events occurring during evening hours that were correlated with high NO and NO$_2$ concentrations were averaged over the length of each single emission plume. In total, 17 emission plumes were observed during the A-LIFE measurement period. Two of these plume events, occurring on April 15 and 16 during wind from northeast, are shown in Figure 7. Clearly, a substantial fraction of the particle size distribution is distributed in the sub-10 nm size region. The mode diameter of the airport emission plumes was inferred by averaging all 17 airport emission events, for which the size distributions are shown in Figure 8 as grey markers. In addition, the events on April 15 and 16 (time series in Figure 7) are depicted in red and blue in Figure 8. The total particle concentration measured by the PSM (light green line in the upper panel of Figure 7) exceeds $10^5$ cm$^{-3}$ during plume-impacted time intervals. All airport emission events are accompanied by elevated NO and NO$_2$ concentrations originating from take-off and taxiway plumes as was e.g. demonstrated by Herndon et al. (2004). The mode diameter of the averaged size distribution of all events (black markers) is at 12.6 nm. Zhu et al. (2011) performed particle





size distribution measurements at similar distances to the emission source at the Los Angeles International Airport and found the particle mode diameter at 12 nm which agrees well with our observation.

## 4 Conclusion

Atmospheric nanoparticle measurements using the newly developed DMA-train with six fixed size channels in the sub-10 nm size range were conducted for the first time during the intensive A-LIFE measurement campaign in April 2017 with the focus on NPF and early particle growth. The complete atmospheric size distribution between 1.8 nm – 10 μm was covered by combined measurements of the DMA-train, the MPSS and APS. The nucleation mode, which typically has the highest number concentrations but is not commonly covered by conventional instrumentation, is added to the number size 285 distribution.

Nucleation mode particles were found to originate from NPF and local pollution. With the nearby airport being the largest local pollution source, we analysed the size distribution of airport emission plumes and found a mean mode diameter at 12.6 nm agreeing with similar studies (Zhu et al., 2011).

Strong particle dynamics at relatively short time scales are revealed in the sub-10 nm size range and three early growth 290 events were characterized using the DMA-train data. Early growth rates were determined and a non-linearity of the particle growth in the sub-10 nm size range was found. During 27 total measurement days, 5 and 7 NPF events were observed by the DMA-train and the MPSS, respectively, which contribute to the CCN budget (Gong et al., 2019).

While NPF is not necessarily detected in both systems if particle growth did not exceed the 10 nm threshold, the DMA train revealed firm insight to sub-10 nm particle evolution. The clear appearance of a new mode followed by growth of the 295 particles below 10 nm was found to be interrupted abruptly by changes in the meteorological conditions. Hence, nanoparticle formation may remain undiscovered if using instrumentation with a lower detection limit of 10 nm. Our data suggest that NPF occurs more frequently than is assumed when particle size distribution measurements do not cover the 1 – 10 nm size range. Extended measurements are needed at high time resolution to fully resolve the particle dynamics in the lower nanometre size range in different environments.


*Data availability.* The data will be available upon request from Sophia Brilke (sophia.brilke@univie.ac.at).

*Author contribution.* B.W. coordinated the A-LIFE field experiment. S.B., N.F., T.M., K.K., and X.G. performed the measurements. S.B., T.M., K.K. and X.G. analysed the data. S.B. wrote the manuscript. S.B., N.F., T.M., K.K., X.G., J.P., 305 B.W. and P.M.W. commented on the manuscript.

*Competing interests.* The authors declare that they have no conflict of interest.





*Acknowledgements.* The authors thank Thomas Ryerson (NOAA Earth System Research Laboratory, Chemical Science
Division, Boulder, CO 80305, USA) for providing the SO₂ instrument and Umar Javed and Anywhere Tsokankunku (Max
Planck Institute for Chemistry, Mainz, Germany) for the calibration of the NO instruments.
This work was supported by the European Research Council under the European Community's Seventh Framework
Programme (FP7/2007/2013)/ERC grant agreement no. 616075. This project has received funding from the ERC under the
European Union's Horizon 2020 research and innovation programme under grant agreement No. 640458 (A-LIFE). K. K.
was funded by the Deutsche Forschungsgemeinschaft (DFG, German Research Foundation) – 264907654; 378741973;
416816480.

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



**Tables and Figures**

**Table 1: Results of the event classification during the measurement period from April 3 – 30, 2017. Those days where an event day was classified either in the DMA-train or in the MPSS are displayed. Growth rates from the DMA-train data in two size intervals (1.8 – 3.2 nm and 3.2 – 10 nm) and from MPSS data in one size interval (10 – 25 nm) are listed here with the prevailing wind**
**direction during the growth in brackets. On April 5, 2017, the growth rate analysis was impossible due to insufficient DMA-train data availability during the event (N/A*). Strong fluctuations in the number size distribution did not allow to infer the growth rate on April 12 (N/A**). The mean condensation sink (CS) was averaged over the course of the event period and is added here.**

| Day of April | DMA-train | MPSS | $GR_{1.8-3.2}$ (nm h$^{-1}$) | $GR_{3.2-10}$ (nm h$^{-1}$) | $GR_{10-25}$ (nm h$^{-1}$) | CS (10$^{-3}$ s$^{-1}$) |
|---|---|---|---|---|---|---|
| 4 | N/A | event | | | 11.0 ± 0.6 [SE] | 7.6 |
| 5 | event | event | N/A* | N/A* | 2.4 ± 0.1 [SE] | 7.0 |
| 8 | non-event | event | | | 5.8 ± 0.7 [NW] | 3.8 |
| 9 | non-event | event | | | 3.5 ± 0.1 [NW] | 6.3 |
| 12 | event | event | N/A** [SE] | N/A** [SE] | N/A** [SE] | 3.6 |
| 16 | event | non-event | 18.2 ± 0.5 [NE] | 5.5 ± 0.1 [SE] | | 7.8 |
| 20 | event | non-event | | 5.8 ± 0.7 [SE] | | 3.5 |
| 21 | undefined | event | | | 2.7 ± 0.1 [SE] | 4.5 |
| 22 | event | event | | 5.7 ± 0.2 [NE] | 3.0 ± 0.1 [NW] | 5.8 |





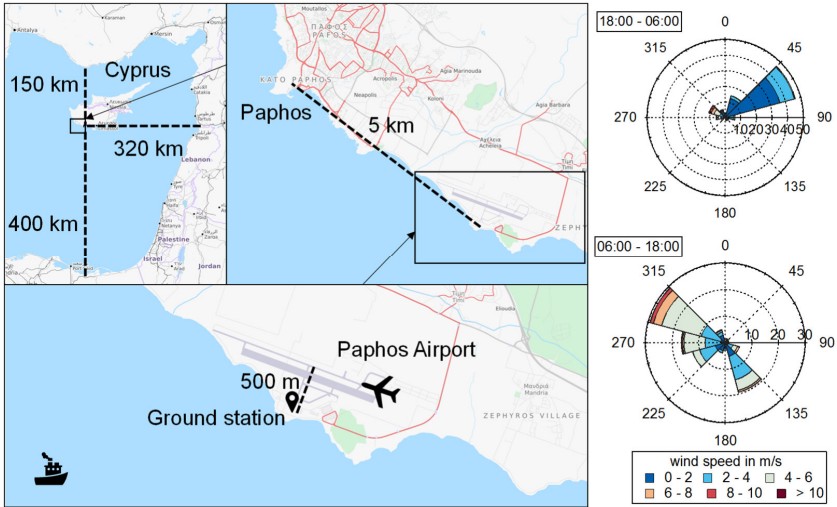


**Figure 1:** **The island of Cyprus is located in the Eastern Mediterranean Sea. Paphos is situated 150 km from the Turkish coast, 320 km from Syria and Lebanon and 400 km from Egypt. The measurement station was located at the coastline at the Paphos International Airport which is 5 km southeast from Paphos city center. The runway of the Paphos airport is at approximately**
**500 m distance northeast to the measurement station. The wind roses in the right panel show the wind frequency distribution during the measurement period, April 1 – 30, 2017. The local land-sea-breeze system dominates the local wind pattern. During night-time (18:00 – 08:00), the wind is primarily northeast at low wind speed between 0.5 – 4 m/s. The sea breeze during daytime (08:00 – 18:00) is mostly southeast or northwest with highest wind speeds. Wind roses were plotted using the Zefir tool (Petit et al., 2017) and the map is from OpenStreetMap (© OpenStreetMap contributors 2019. Distributed under a Creative Commons BY-SA**
**License.).**



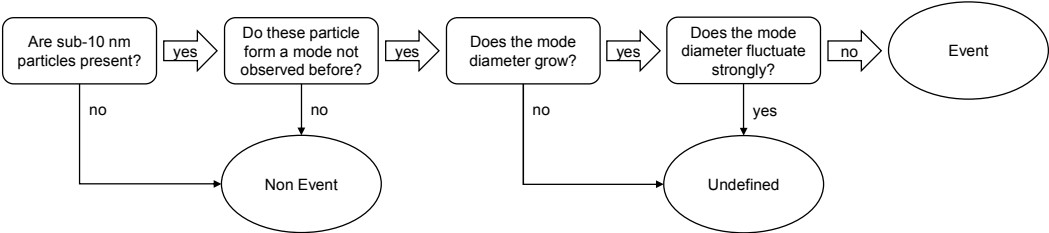

**Figure 2: Classification scheme modified from Dal Maso et al. (2005) for categorizing the sub-10 nm DMA-train data into event, non-event and undefined days.**





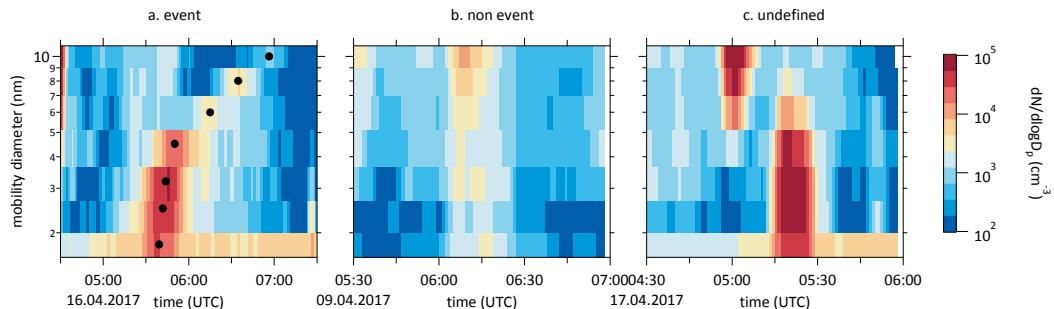

**Figure 3: Three different patterns of the sub-10 nm particle number size distribution measured with the DMA-train. The left panel (a.) shows the case of an event day exhibiting a growing mode diameter (black markers). A non-event day is illustrated in the middle panel (b.) where no new particle mode is formed. The right panel (c.) shows an undefined day with a new particle mode forming which does not show signs of particle growth.**




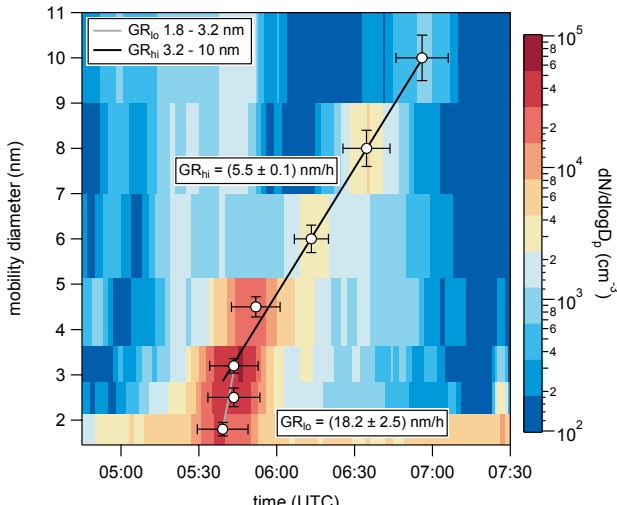


**Figure 4: Event on April 16, 2017. The set mobility diameter in each size channel is plotted versus the time of the mode peak (white markers) with the time series of the number size distribution in the background. The growth rate can be directly determined from the slope of the fit to the diameter by applying a linear fit using an orthogonal distance regression. DMA-train growth rates are**
**typically divided into two size intervals which also applies to this ambient data set. In this example from April 16, 2017, an initial growth rate of 18.2 nm/h in the lower size interval (1.8 – 3.2 nm) is calculated. The growth rate in the higher size interval (3.2 – 10 nm) is 5.5 nm/h. Note the linear scale on the y-axis.**



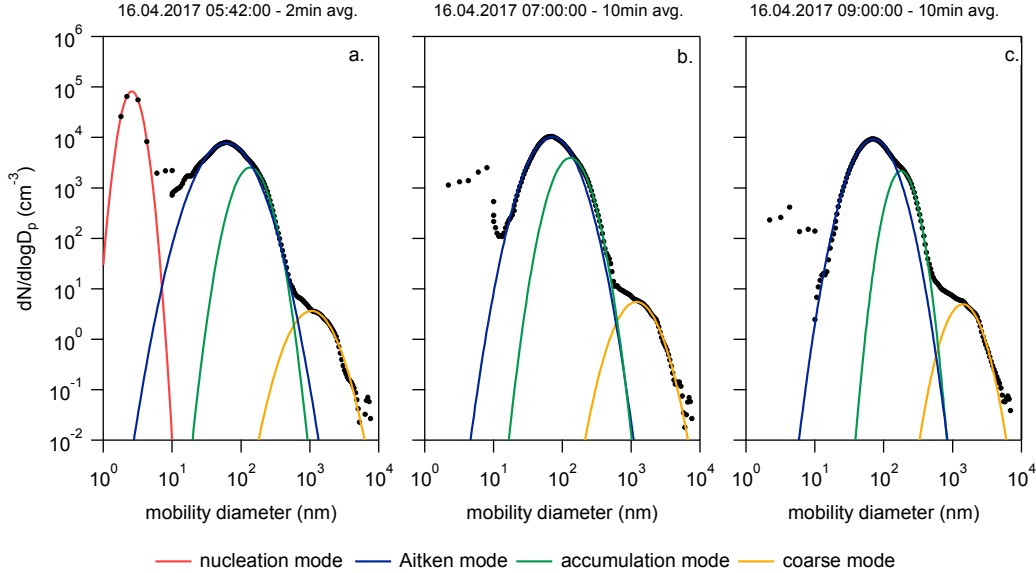


**Figure 5: Averaged full number size distribution DMA-train data resolving particle diameters < 10 nm, MPSS data (>10 nm) and APS data (>800 nm). Nucleation mode, Aitken mode, accumulation mode and coarse mode were fitted using a log-normal function. Panel a. shows the size distribution during the NPF event on April 16, 2017, with a clear nucleation mode appearing which is reduced at 07:00, as shown in panel b. In panel c., the size distribution after the NPF is demonstrated. Uncertainties on the number**
**concentration are estimated to 20% for diameters < 10 nm, 30% between 10 and 20 nm and 10% for diameters beyond 20 nm.**



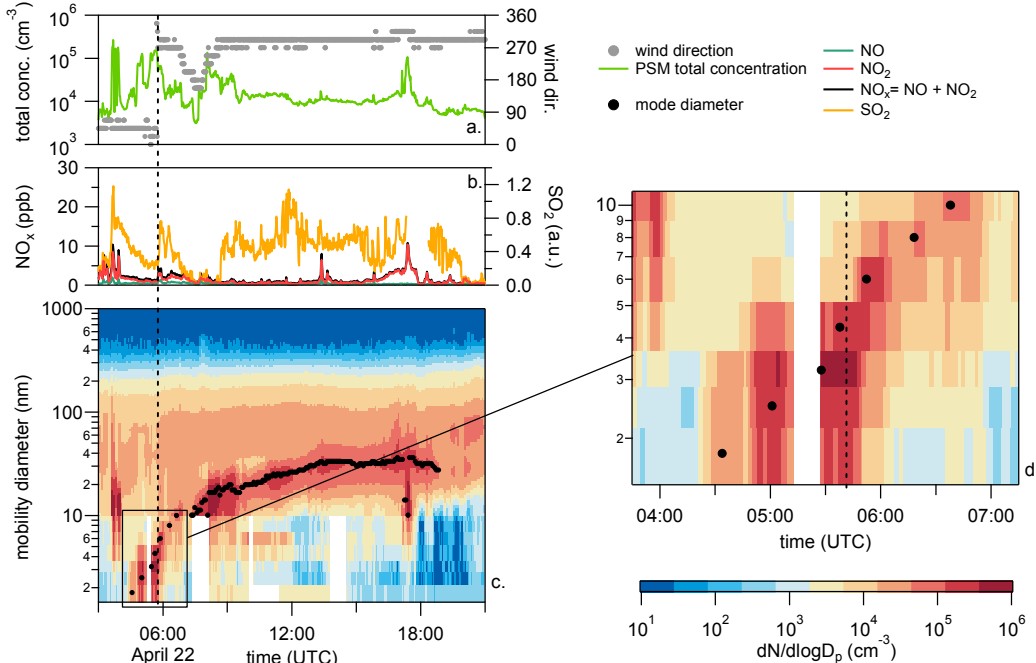

**Figure 6: NPF event on April 22, 2017. Panel a. shows the 1-min averaged time series of the total particle concentration measured by the PSM on a logarithmic scale and the wind direction (grey markers). Trace gases (NO, NO₂, NOₓ=NO+NO₂) are shown in panel b. in blue, red and black. The qualitative SO₂ data was added in orange. In panel c., the size distribution from DMA-train and MPSS data during the complete NPF event is shown. Panel d. depicts the close-up of the start of the NPF event that exhibits bursts of particles in two stages before continuous particle growth takes off. The mode diameter is shown as black markers and the black dashed line indicates the change in wind direction (grey markers in upper panel).**



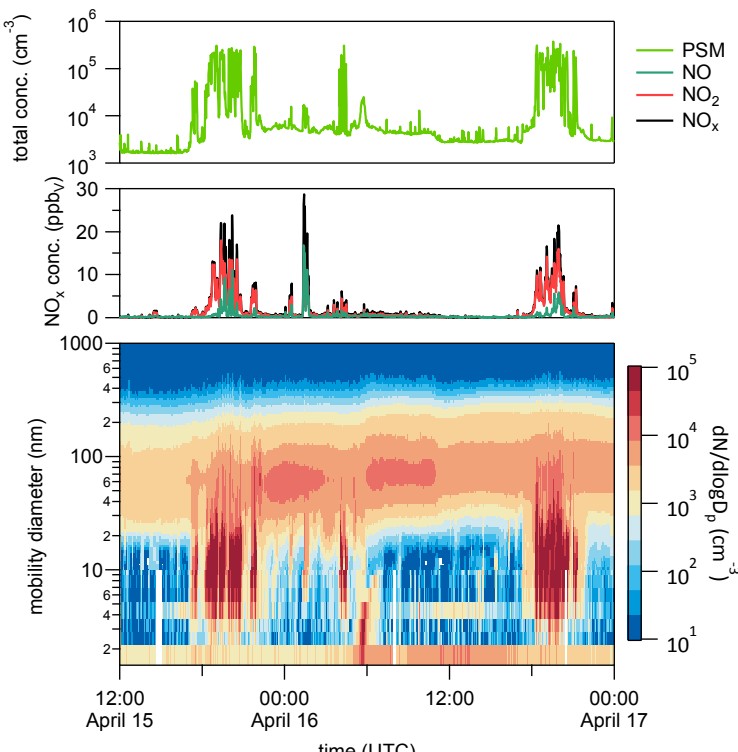

**Figure 7: Time series of the particle size distribution from 1 – 1000 nm (lower panel) during airport emission plumes. The upper panel shows the 1-min averaged time series of the total particle concentration measured by the Airmodus A10 Particle Size Magnifier (PSM) on a logarithmic scale. Trace gases (NO, NO₂, NOₓ=NO+NO₂) are shown in the middle panel. Airport emissions can clearly be assigned to the plumes of nucleation mode particles being carried to the station during N-NW wind situations (wind not shown here). The particle plumes with maximum particle concentrations of $3 \cdot 10^5 \, cm^{-1}$ (upper panel) are accompanied by elevated NO/NO₂ levels (middle panel). On April 16, a new mode appears in the sub-10 nm size range showing well the different behaviour between emission plumes and NPF.**



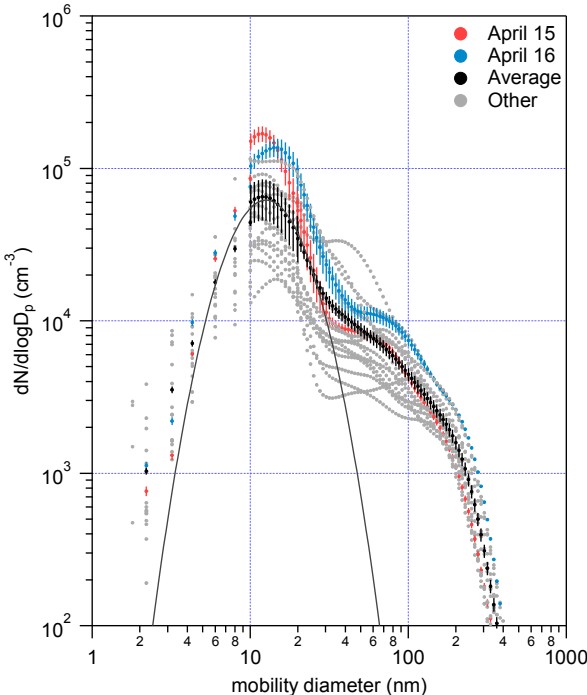

**Figure 8: Number size distributions during high airport emission loads. The 2h averaged size distributions of the time series presented in Figure 7 shows particle plumes on April 15 (red) and April 16 (blue) yielding high particle number concentrations and elevated NO/NO₂ concentrations. Other airport emission plumes during the measurement period are shown in grey. All number size distributions were averaged (black). The distribution was fitted using a log-normal function (dark grey line) and the resulting mode diameter is at 12.6 nm.**

525

530