# Peer review of "New Particle Formation and Sub-10 nm Size Distribution Measurements during the A-LIFE field experiment in Paphos, Cyprus"

_Atmospheric Chemistry and Physics, 2019_

## Referee Comment (RC1) · Anonymous Referee #1 · 15 Jan 2020

General Comments

This work presents the results from a field campaign in Eastern Mediterranean, focusing on new particle formation and presenting results from a novel instrument, the DMA-train. The deployment of such new techniques to the field is expected to provide valuable information on new particle formation processes. Nevertheless, improvements both in the presentation and the content are required so that this manuscript meets the standards of ACP.

As the DMA-train is a new instrument and no field measurements demonstrating its capabilities are available I strongly recommend that there is a dedicated paragraph

showing all available measurements in terms of total and segregated channel number concentrations, correlations between various channels, field calibration performance if available, comparison to other instruments (MPSS,PSM), formation rates.

With respect to the campaign itself there is sufficient room for discussion of all events, even if it comes to a few sentences. Air mas back trajectories, meteorological conditions and additional measurements should be described.

Specific Comments

Abstract: Please consider changing this paragraph to Past tense, past and present are used simultaneously ie "The newly developed DMA-train is deployed" and "The nearby Paphos airport was found to be". The same stands for the rest of the manuscript.

Line 54: There should be also a reference to Particle Size Magnifier (PSM).

Line 72: Update to Kalivitis et al., 2019.

Line 99: The instrument types for the trace gas measurements should be described.

Line 121: The flow rate of 5th and 6th CPCs are not mentioned.

Line 143: It is not clear to me why the overall uncertainty is set to 20%, please explain further.

Line 148: What about corrections for diffusional loses in the tubing and within the DMAs? These loses are significant for the nucleation size ranges and they have definitely to be taken into account.

Line 150: The formation rates calculated form the DMA-train should also be discussed in the manuscript.

Line 164: To my opinion Figures S2 (and hence Figure 4) and S3 should be merged in a single Figure and presented in the manuscript, as they are totally in line with the description in the main text and they will assist the reader understand the methodology

for calculating GRs. On the other hand, the scheme by Dal Maso et al., (2005) in Figure 2 is well established and used repeatedly in the literature and there is no need to present it again.

Line 183: The number of events should be clearly stated rather than "several".

Line 190: During the study period there was only one event that according to Table 1 was captured simultaneously from the DMA-train and the MPSS, or at least it was the only event that GR in size ranges from both techniques was possible to be calculated (22 April). How many events were observed by both techniques, it is not clear form Table 1.

Line 212: It is worth presenting the number concentrations of different size ranges of aerosol particles during the whole study period. At least for the nucleation mode, and since there were strong pollution sources, it would be interesting to see how various process contribute to the number concentrations.

Line 220: There is no discussion here about trace gases and only little discussion about meteorological parameters. What were the wind velocities, temperature range, cloud cover, trace gases, PM etc. A more thorough discussion of the event is required.

Line 240: In panel c. there are nucleation mode particles, even if not as a clear mode. Dilution is a possible explanation, or perhaps scavenging. It is also recommended that the survival probability is calculated.

Figure 1: In the wind rose diagrams the hours in the frame are 18:00-06:00 and 06:00 -18:00, while in the caption the times are 18:00-08:00 and 08:00-18:00. Correct to the proper time period and additionally state if it is local or UTC. Are the displayed data hourly averaged, please explain what the radial frequency is.

Figure 3: The middle panel (b.) depicts an undefined event rather than a non event. Prior to 06:00 there are no nucleation mode particles, while after that significant concentrations are observed but no growth of these particles. To my opinion it is closer to

c. rather to a non-event.

References

Kalivitis et al., https://doi.org/10.5194/acp-19-2671-2019, 2019

---

## Referee Comment (RC2) · Santtu Mikkonen (Referee) · 9 Feb 2020

General comments

This paper introduces results from measurement campaign made in Paphos, Cyprus. The main goal of the campaign was to show how the novel DMA-train setup performs in field measurements. The article is nicely written and introduces few interesting looking cases of new particle formation events. DMA-train has been shown to work in controlled conditions in CLOUD measurements and this manuscript illustrates that the system is also applicable outside of the lab. However, as the novelty in this manuscript is in instrumental development I feel that Atmospheric measurement techniques might

be better target journal for this, but I leave this under editor's consideration.

Major comment

As the reviewer 1 already noted, this manuscript needs lots of additional information on the conditions at the measurement site. Multiple papers have shown that local meteorology, trace gas concentrations and long-range transport affect greatly on new particle formation and thus information on these should be provided. Especially if the paper should be published in ACP. In AMT technical details would be more important.

Minor comments

in page 6 line 162: I was happy to see that the fit was made with orthogonal distances instead of ordinary least squares as the OLS would give too low estimates for these type of data with high uncertainties. Did you consider other methods taking account the uncertainties?

Page 9 lines 246-264 and Fig 6: This section would benefit of more quantitative approach in discussing the relationships between particle concentrations and trace gases. Fig 6 does not really give information on the trace gas concentrations as the y-axis for NOx is poorly chosen. I understand the consistency with fig 7 but here it loses the information in the figure.

---

## Author Comment (AC1) · 23 Mar 2020

Author's response to Referee # 1

We thank the anonymous Referee # 1 for valuable comments on the manuscript and suggestions for improvement.

—

General comments (1) This work presents the results from a field campaign in Eastern Mediterranean, focusing on new particle formation and presenting results from a novel instrument, the DMA-train. The deployment of such new techniques to the field

is expected to provide valuable information on new particle formation processes. Nevertheless, improvements both in the presentation and the content are required so that this manuscript meets the standards of ACP. As the DMA-train is a new instrument and no field measurements demonstrating its capabilities are available I strongly recommend that there is a dedicated paragraph showing all available measurements in terms of total and segregated channel number concentrations, correlations between various channels, field calibration performance if available, comparison to other instruments (MPSS,PSM), formation rates. With respect to the campaign itself there is sufficient room for discussion of all events, even if it comes to a few sentences. Air mas back trajectories, meteorological conditions and additional measurements should be described.

(2) Our study constitutes the first deployment of the DMA-train in an atmospheric experiment. However, previously published articles have presented results on the technical background on DMA-train measurements in (Stolzenburg et al., 2017) and the performance of the DMA-train during CLOUD chamber experiments Stolzenburg et al. (2018). In the study of Stolzenburg et al. (2017), the capability of the DMA-train to act as a size spectrometer was already demonstrated showing the single size channels during a chamber nucleation event. The sequential appearance of the fixed size channels characterizing an NPF event is shown here again in Fig. S2. Gong (2019) published their results in ACP last year and provided an overview of the available ground-based measurements during A-LIFE regarding trace gases, MPSS data, total number concentration and meteorological and air quality parameters. Their study includes a detailed description on the setup of the instrumentation at the measurement site as well as a time series of the particle number size distribution of the MPSS data and total number concentration throughout the whole campaign. Correlation of NOx and total number concentration are shown in the SI of Gong (2019). We have discussed the possibility to include air mass trajectories in the analysis within the co-authors, however, we have decided against it. The measurement site is characterized with strong local pollution sources, therefore we think that the consideration of large scale trajectories should not to be applied in this specific case. The following changes will be made in the manuscript to account for the referee's suggestions:

(3) Line 107: A detailed overview of the meteorological parameters and air quality parameters during the measurement campaign is provided in Gong et al. (2019). Line 221: The wind and trace gas data were analysed to distinguish between NPF and nucleation mode peaks from plume impact. First signs of NPF were observed during morning hours after sunrise typically during clear-sky conditions at low wind speed (< 4 m/s) and at values below 3 ppbv NOx. During plume-related peaks, NOx values reached values as high as 40 ppbv.

Specific comments: (1) Abstract: Please consider changing this paragraph to Past tense, past and present are used simultaneously ie "The newly developed DMA-train is deployed" and "The nearby Paphos airport was found to be". The same stands for the rest of the manuscript.

(2) We will make sure that the tense is correctly chosen in the abstract and the rest of the manuscript.

(1) Line 54: There should be also a reference to Particle Size Magnifier (PSM).

(2) Yes, we will also include the turbulent-mixing type PSM in this enumeration.

(3) Line 55: By implementing a low cut-off CPC, e.g. a diethylene glycol-based laminar-flow CPC or turbulent mixing-type Particle Size Magnifier (PSM) (Vanhanen et al., 2011), the complete number size distribution of neutral particles has been measured during nucleation events in the atmosphere (Jiang et al., 2011)

(1) Line 72: Update to Kalivitis et al., 2019.

(2) This reference will be updated.

(1) Line 99: The instrument types for the trace gas measurements should be described

(2) Thanks for pointing this out, the instrument types will be described here.

(3) Line 102: Complementary trace gas measurements were performed using a NO-NO2 monitor (Advanced Pollution Instruments API 200), O3 monitor (Thermo Scientific Model 49i) and SO2 monitor (Thermo Environmental Instruments Model 43C Trace Level Pulsed Fluorescence Sulfur Dioxide Analyzer) with periodic background measurements performed using synthetic air 5.0.

(1) Line 121: The flow rate of 5th and 6th CPCs are not mentioned.

(2) We will include the flow rates.

(3) Line 126: A water-based TSI CPC Model 3788 was employed in the fifth channel at 1.5 L/min inlet flow rate. The fifth and sixth channel were operated with a TSI UCPC Model 3776 at standard temperature settings (see Table S1 in SI). Here, the DMA voltage was not entirely fixed but alternated in a 10 s time interval to cover one additional size channel at 1.5 L/min inlet flow rate.

(1) Line 143: It is not clear to me why the overall uncertainty is set to 20%, please explain further.

(2) The overall uncertainty is set to 20% as a conservative best estimate. The uncertainty is estimated from the uncertainties on sampling, charging, DMA transmission and CPC detection efficiency. The uncertainty on CPC detection efficiency in the sub-10 nm size range is assumed to make the largest contribution to the overall uncertainty ($\sim$ 10 %). We will clarify this in the updated manuscript as follows:

(3) Line 149: As an overall conservative best estimate considering uncertainties from diffusional losses, DMA transmission and charging efficiency, we assume $\pm$20% uncertainty on the number concentration in each size channel.

(1) Line 148: What about corrections for diffusional loses in the tubing and within the DMAs? These losses are significant for the nucleation size ranges and they have definitely to be taken into account.

(2) As pointed out by the referee #1 the diffusional losses in the tubing and within the

DMAs are important to be considered. The DMAs were characterized in the technical DMA-train paper Stolzenburg (2017). In this analysis both diffusional losses and DMA transmission were considered, and we will clarify this in the updated manuscript.

(3) Line 137: Corrections were applied to account for diffusional particle losses in the sampling lines and transmission in the DMAs.

(1) Line 150: The formation rates calculated form the DMA-train should also be discussed in the manuscript.

(2) The formation rate of particles is indeed an important parameter to describe the particle dynamics. Unfortunately, in this study, we observed only one event on April 16 starting in the lowest DMA-train channel. The estimated formation rate of 1.8 nm particles calculated from the DMA-train data lies on the order of 10 cm-3 s-1 in this single specific case. However, with a larger dataset more detailed information on the particle formation rate may be collected in future studies.

(1) Line 164: To my opinion Figures S2 (and hence Figure 4) and S3 should be merged in a single Figure and presented in the manuscript, as they are totally in line with the description in the main text and they will assist the reader understand the methodology for calculating GRs. On the other hand, the scheme by Dal Maso et al., (2005) in Figure 2 is well established and used repeatedly in the literature and there is no need to present it again.

(2) We put these figures on purpose in the supplemental information since they add extra information on the methodology and not to our findings. Figure 2 shows a scheme for the classification of sub-10 nm particles, however, this scheme is very well established as pointed out by referee #1 and Figure 2 will therefore be removed from the updated manuscript and the main text will be changed correspondingly.

(1) Line 183: The number of events should be clearly stated rather than "several".

(2) We will be more precise here.

(3) Line 190: Five events were classified based on the DMA-train data in the sub-10 nm size regime.

(1) Line 190: During the study period there was only one event that according to Table 1 was captured simultaneously from the DMA-train and the MPSS, or at least it was the only event that GR in size ranges from both techniques was possible to be calculated (22 April). How many events were observed by both techniques, it is not clear form Table 1.

(2) We will clarify this by adding the corresponding instrument names to the size ranges given in line 198:

(3) Line 198: For the sub-10 nm size distribution measured by the DMA-train, an event was identified in 5 cases whereas 7 event days were found for particles larger than 10 nm (MPSS) during 27 total measurement days.

(1) Line 212: It is worth presenting the number concentrations of different size ranges of aerosol particles during the whole study period. At least for the nucleation mode, and since there were strong pollution sources, it would be interesting to see how various process contribute to the number concentrations.

(2) The number concentrations at different particle sizes is in principle presented in the particle number size distribution (PNSD). In Gong (2019), the PNSD during polluted (airport-affected) and unpolluted time periods were already presented. In our study we have added the PNSD including the DMA-train data during airport emission plumes (Fig. 8). The other case, when nucleation mode particle number concentrations are higher, is when NPF occurs, as is shown in Figure 5.

(1) Line 220: There is no discussion here about trace gases and only little discussion about meteorological parameters. What were the wind velocities, temperature range, cloud cover, trace gases, PM etc. A more thorough discussion of the event is required.

(2) Please find our comments on these recommendation in the general comment as

well as the suggested changes to the manuscript.

(1) Line 240: In panel c. there are nucleation mode particles, even if not as a clear mode. Dilution is a possible explanation, or perhaps scavenging. It is also recommended that the survival probability is calculated.

(2) We have adapted line 237 as follows:

(3) Line 237: A new air mass being carried to the measurement station might lead to dilution of the condensable vapours participating in the sub-10 nm growth or increased scavenging to larger particles.

(1) Figure 1: In the wind rose diagrams the hours in the frame are 18:00-06:00 and 06:00-18:00, while in the caption the times are 18:00-08:00 and 08:00-18:00. Correct to the proper time period and additionally state if it is local or UTC. Are the displayed data hourly averaged, please explain what the radial frequency is.

(2) We will correct this error in the figure caption.

(1) Figure 3: The middle panel (b.) depicts an undefined event rather than a non event. Prior to 06:00 there are no nucleation mode particles, while after that significant concentrations are observed but no growth of these particles. To my opinion it is closer to c. rather to a non-event.

(2) Panel b. in Figure 3 was declared as non-event since no growth was observed. However, as stated by referee#1, the increase in particles at diameters can also be classified as a new mode being formed. However, to add more information to this plot, we would like to present two event cases (a. and b.) of which the second case shows growth beyond 10 nm. Please see added Figure:

(3) Fig. caption: A second event case is illustrated in the middle panel (b.) where a new particle mode is formed and continues to grow beyond 10 nm.

———————————————

Author's response to Referee # 2

We thank Santtu Mikkonen for reviewing our manuscript and adding some important comments for improvement on the manuscript.

—

Major comment (1) This paper introduces results from measurement campaign made in Paphos, Cyprus. The main goal of the campaign was to show how the novel DMA-train setup performs in field measurements. The article is nicely written and introduces few interesting looking cases of new particle formation events. DMA-train has been shown to work in controlled conditions in CLOUD measurements and this manuscript illustrates that the system is also applicable outside of the lab. However, as the novelty in this manuscript is in instrumental development I feel that Atmospheric measurement techniques might be better target journal for this, but I leave this under editor's consideration. As the reviewer 1 already noted, this manuscript needs lots of additional information on the conditions at the measurement site. Multiple papers have shown that local meteorology, trace gas concentrations and long-range transport affect greatly on new particle formation and thus information on these should be provided. Especially if the paper should be published in ACP. In AMT technical details would be more important.

(2) As already stated in the answers to referee #1, we will add an explanatory sentence and refer to the study of Gong et al. where the meteorological conditions at the measurement site have been described in detail. Regarding the long-range transport, we have decided against including backward trajectories in our analysis since this measurement site is characterized with strong local pollution sources – information from backward trajectories may thus lead to misinterpretations.

Minor comments (1) in page 6 line 162: I was happy to see that the fit was made with orthogonal distances instead of ordinary least squares as the OLS would give too low estimates for these type of data with high uncertainties. Did you consider other

methods taking account the uncertainties?

(2) This is our standard procedure to calculate DMA-train growth rate and other methods were not considered in this analysis.

(1) Page 9 lines 246-264 and Fig 6: This section would benefit of more quantitative approach in discussing the relationships between particle concentrations and trace gases. Fig 6 does not really give information on the trace gas concentrations as the y-axis for NOx is poorly chosen. I understand the consistency with fig 7 but here it loses the information in the figure.

(2) The relationship between particle concentrations and NOx was discussed in Gong (2019) SI, we put a reference to this paper here. The referee is correct that Figure 6 loses a bit in information by the x-axis setting, we will adapt the y-axis.

————————————————

Gong, X.: Supplement of Characterization of aerosol properties at Cyprus , focusing on cloud con- densation nuclei and ice-nucleating particles, 2019.

Stolzenburg, D., Steiner, G. and Winkler, P. M.: A DMA-Train for precision measurement of sub-10nm aerosol dynamics, Atmos. Meas. Tech., 10, 1639–1651, doi:10.5194/amt-10-1639-2017, 2017.

Stolzenburg, D., et al.: Rapid growth of organic aerosol nanoparticles over a wide tropospheric temperature range, Proc. Natl. Acad. Sci. U. S. A., 115(37), 9122–9127, doi:10.1073/pnas.1807604115, 2018.

————————————————————

[Figure]

[Figure]

**Fig. 1.** Three different patterns of the sub-10 nm particle number size distribution.

**Fig. 2.** NPF event on April 22, 2017.